The influences of N-acetyl cysteine (NAC) on the cytotoxicity and mechanical properties of Poly-methylmethacrylate (PMMA)-based dental resin

Jiao Yang 1 4
Ma Sai 1 4
Li Jing 2
Shan Lequn 3
Yang Yanwei 1
Li Meng 1
Chen Jihua 1 jhchen@fmmu.edu.cn
1 State Key Laboratory of Military Stomatology, Department of Prosthodontics, School of Stomatology, the Fourth Military Medical University , Xi’an , PR China
2 Department of Orthopaedic Oncology, Xijing Hospital Affiliated to the Fourth Military Medical University , Xi’an , PR China
3 Department of Orthopaedic Surgery, Tangdu hospital, the Fourth Military Medical University , Xi’an , PR China
Grant Melissa
4 Co-first author

Electronic publication date: 2015 Apr 23
Publication date: 2015
Volume: 3
Electronic Location ID: e868
Received 2015 Jan 6; Accepted 2015 Mar 10
Copyright: © 2015 Jiao et al.
Copyright year: 2015
Copyright holder: Jiao et al.
License: This is an open access article distributed under the terms of the Creative Commons Attribution License, which permits unrestricted use, distribution, reproduction and adaptation in any medium and for any purpose provided that it is properly attributed. For attribution, the original author(s), title, publication source (PeerJ) and either DOI or URL of the article must be cited.
License URL: https://creativecommons.org/licenses/by/4.0/

Keywords: Poly-methylmethacrylate (PMMA), N-acetyl cysteine (NAC), Cell viability, Mechanical properties, Detoxification

Funding: National Nature Science Foundation of China 81130078 81300927 IRT13051 This study was financially supported by grant 81130078 (principal investigator Jihua Chen) and grant 81300927 (principal investigator Sai Ma) from the National Nature Science Foundation of China, and Program No. IRT13051 from the Program for Changjiang Scholars and Innovative Research Team in University (PCSIRT). The funders had no role in study design, data collection and analysis, decision to publish, or preparation of the manuscript.

==============================
Objectives. This study aimed to investigate the influences of N-acetyl cysteine (NAC) on cytotoxicity and mechanical properties of Poly-methylmethacrylate (PMMA) dental resins.

Methods. Experimental PMMA resin was prepared by incorporating various concentrations of NAC (0, 0.15, 0.3, 0.6 and 0.9 wt.%). MTT assay was performed to investigate viability of human dental pulp cells after exposure to extract of PMMA resin with or without NAC. Cell adhesion on resin specimens was examined with scanning electron microscopy. Degree of conversion was studied with Fourier Transform Infrared Spectroscopy (FTIR). Flexural strength, microhardness and surface roughness was evaluated using a universal testing machine, microhardness tester and optical profilometer, respectively.

Results. Incorporation of NAC into PMMA resin significantly reduced its cytotoxicity and enhanced cell adhesion on its surface. NAC induced negative influences on the mechanical and physical properties of PMMA resin in a dose-dependent manner. The degree of conversion for all experimental PMMA resins reached as high as 72% after 24 h of polymerization. All the tested properties were maintained when the concentration of incorporated NAC was 0.15 wt.%.

Conclusion. The addition of 0.15 wt.% NAC remarkably improved biocompatibility of PMMA resin without exerting significant negative influence on its mechanical and physical properties.

Introduction

Self-polymerizing poly-methylmethacrylate (PMMA)-based dental resin, consisting of a polymer powder and a liquid monomer, is one of the most frequently-used materials in dentistry (Burns et al., 2003). It can be used for fabrication of temporary crowns, denture repair, and the temporary seal of prepared cavities. There have been concerns that when PMMA resin is loaded directly onto prepared teeth, such as in the case of temporary crown fabrication, unpolymerized monomers released from the resin may invade into the dental pulp through dentinal tubules and thus induce negative influences on the pulp tissue (Inoue, Miyakoshi & Shimono, 2001; Noda et al., 2002).

The cytotoxic effects of various dental monomers have been well-documented. Although the detailed mechanism is still largely unknown, there is striking evidence indicating that the disturbance of intracellular redox balance is related to monomer-related cytotoxicity (Ciapetti et al., 2002; Huang & Chang, 2005; Ratanasathien et al., 1995; Vale et al., 1997). It has been reported that dental monomers reduced the intracellular levels of glutathione (GSH), an important non-enzymatic anti-oxidant that protects the cell against oxidative stress, and thus induces the over-production of reactive oxygen species (ROS). Such disturbance of cellular redox balance would eventually result in cell function abnormality and even cell death (Bakopoulou, Papadopoulos & Garefis, 2009; Schweikl, Spagnuolo & Schmalz, 2006).

Recently, N-Acetyl Cysteine (NAC), a well-known antioxidant, has been found effective in protecting cells against dental monomer-related cytotoxicity (Lee et al., 2006; Paranjpe et al., 2008a; Paranjpe et al., 2008b; Schweikl et al., 2007; Spagnuolo et al., 2006). When incorporated in PMMA resins, NAC restored cell viability and function to a biologically significant degree (Att et al., 2009; Kojima et al., 2008; Yamada et al., 2009; Yamada et al., 2008). Considering that mechanical properties are also important factors that determine the clinical success of PMMA resins, it is necessary to study the influences of NAC on various mechanical and physical properties of PMMA resins. However, up to now, there is limited published data regarding this aspect. Although Att et al. (2009) demonstrated that the addition of NAC up to 0.6 wt.% induced no significant negative influences on the transverse strength and elastic modulus of PMMA dental resin, more studies needs to be performed to investigate the influences of NAC on other important mechanical properties, such as degree of conversion (DC), microhardness, surface roughness, of PMMA dental resins.

Thus, the purposes of the present study are: first, to confirm the protective effects of NAC against the cytotoxicity of PMMA dental resin; second and more importantly, to investigate the influences of NAC on various mechanical and physical properties of PMMA dental resin.

Materials and Methods

Resin preparation

Self-polymerizing PMMA resin (Unifast Trad, GC, Tokyo, Japan) was prepared by mixing the powder and liquid components for 30 s in accordance with the manufacturer’s recommendations (powder/liquid ratio of 1.0 g/0.5 mL) in a 12-well culture plate. NAC (Sigma; St Louis, MO, USA) was prepared as 1.0 mol/L stock solution in HEPES buffer (Sigma; St Louis, MO, USA), and the pH was adjusted to 7.2 as previously reported (Yamada et al., 2008). To prepare NAC-loaded PMMA resin (0.15, 0.3, 0.6, 0.9 wt%), 13.8, 27.6, 55.2 or 82.8 µL NAC solution was added to the liquid MMA to a final volume of 0.5 mL, and then this NAC-incorporating liquid was mixed with 1.0 g powder. After 30 min of setting, the polymerized resin block was rinsed with ddH2O once.

Preparation of the extract

The extract of the polymerized resin was prepared by soaking PMMA resin specimens with or without NAC in alpha-modified Eagle’s medium (α-MEM; Gibco BRL Division of Invitrogen, Gaithersburg, Maryland, USA) without serum in the 12-well culture plate at 37 °C in a humid atmosphere of 5% CO2 for 24 h. According to ISO 10993-12:2007, the ratio between the surface of the sample and the volume of the added medium was 1.25 cm2/mL. The eluent was filtered for sterilization and 10% fetal bovine serum (Gibco, Gaithersburg, Maryland, USA) was added. The extract was stored at 4 °C before use.

Quantification of released NAC with high performance liquid chromatography (HPLC)

HPLC analysis was performed to determine the concentration of released NAC from the experimental PMMA resin containing 0.9 wt% of NAC. The extract was prepared as previously described. HPLC analysis (Model Shimadzu LC-2010A; Shimadzu Corporation, Kyoto, Japan) was performed at 220 nm on an InertSustain C18 column (5 µm, 4.6 mm × 250 mm; GL Sciences, Shinjuku-ku, Tokyo, Japan). The column was maintained at 50 °C and the samples were eluted for 5 min. The mobile phase, at a flow rate of 0.7 mL/min, consisted of methanol and 0.2% phosphoric acid with 100 mM sodium perchlorate. The volume of sample injected was 20 µL. Identification of the analyte, NAC, was made based on the retention time of the NAC peaks registered for the standard solutions. Standard curve was drawn using NAC solution with various concentrations (1, 5, 20, 40, 60, 80 and 100 µg/mL). After the linearity of the standard curve was confirmed by linear regression analysis, the concentration of released NAC from the experimental PMMA resin was calculated using the standard curve. The experiments were repeated for three times, and the mean value was determined as the concentration of released NAC.

Cell cultures

Human dental pulp cells (HDPCs) were isolated from dental pulp tissue of non-carious third molars extracted from young healthy patients (18–25 years old) according to a protocol verbally approved by the Ethics Committee of the Fourth Military Medical University. Briefly, the pulp was harvested, minced and digested in a solution containing 3 mg/mL type I collagenase and 4 mg/mL dispase (Gibco, Gaithersburg, Maryland, USA) at 37 °C for 2 h (Hilkens et al., 2013). Single-cell suspensions were obtained by passing the cells through a 70-µm strainer (BD Falcon, Franklin Lakes, NJ, USA). The cells were cultured in α-MEM (Gibco) supplemented with 10% fetal bovine serum (Gibco, Gaithersburg, Maryland, USA), 100 units/mL penicillin and 100 mg/mL streptomycin. The medium was changed every 3 days. Cells from the second passage were used for both cell attachment assay and cytotoxicity assays.

Cell viability test

Mitochondrial dehydrogenase activity was measured by MTT assay to investigate cell viability. HDPCs were seeded into 96-well culture plates at a density of 5 × 103 cells/well and incubated at 37 °C and 5% CO2 for approximately 24 h. When the cells grew to 80% confluence, the medium was replaced by the extract of PMMA resin with or without NAC. After each incubation period (3 days, and 7 days), 20 µL of MTT solution at 5 mg/mL (Sigma-Aldrich, St. Louis, Misouri, USA), disinfected by infiltration, was added to the well, and the plates were incubated for a further 4 h. After disposal of the culture medium, 200 µL dimethyl sulfoxide (DMSO) (Amresco, Solon, Ohio, USA) was added to each well. After gently swirled for 10 min, the absorbance of each well at 490 nm was measured. The blank group contains only complete culture medium without cells. For the control group, cells were cultured in wells without eluent. Relative cell viability was calculated using the following equation: Relative cell viability=ODexperimental−ODblank/ODcontrol−ODblank.

The experiments were repeated for three times. Cytotoxicity was rated in accordance with ISO-standard 10993-5 as non-cytotoxic (cell viability higher than 75%), slightly cytotoxic (cell viability ranging from 50% to 75%), moderately cytotoxic (cell viability ranging from 25% to 50%), and severely cytotoxic (cell viability lower than 25%).

The MTT assay was also employed to evaluate the influence of NAC alone on cells. Using HPLC analysis, we confirmed that the concentration of NAC released from the experimental PMMA resin containing NAC was no higher than 0.54 mM. Thus, to exclude the possibility that released NAC may enhance the proliferation of cells, we used 0.54 mM NAC alone to treat the cells and studied its influence on cell viability with MTT assay.

Cell attachment assay

Disc-shaped specimens (10-mm in diameter and 2-mm in thickness) of each subgroup were fabricated using a mould. The paste was filled in the mould and covered with a glass silde. All the specimens were allowed to set for 30 min at 25 °C. All discs were rinsed with ddH2O once and sterilized by ethylene oxide gas before use. HDPCs were seeded onto the discs at the density of 2.0 × 106 cells/well in a 6-well culture plate. After incubation for 24 h, the samples were rinsed with phosphate-buffered saline (PBS, pH 7.2) to eliminate unattached cells and fixed with 2.5% glutaraldehyde and 2% paraformaldehyde in 0.2 M sodium cacodylate buffer (pH = 7.4). After being freeze-dried and sputter-coated with platinum (E-1030; Hitachi, Tokyo, Japan), the specimens were observed with SEM (Model S-4800; Hitachi, Tokyo, Japan) to examine cell morphology and attachment.

Measurements of degree-of-conversion (DC)

The degree of auto-polymerization conversion of control and experimental resin specimens was measured by Fourier Transform Infrared Spectroscopy (FTIR 8400S; Shimadzu Scientific Instruments, Kyoto, Japan). The resin paste was placed between two polyethylene films and pressed to form a very thin film. The FTIR spectra were collected after 5 min, 10 min, 20 min, 30 min, 1 h, 4 h and 24 h. FTIR spectrum of the uncured mixture was also recorded as reference. For each subgroup, the measurement was repeated for three times.

DC of the resin specimens were calculated using the following equation: DC%=1−Absana/AbsrefcuredAbsana/Absrefuncured×100%.

Absana: the intensity of the carbon–carbon double bond stretching vibration (peak height at 1,638 cm−1);

Absref: the intensity of the symmetric ring stretching at 1,610 cm−1;

Absorbance peak intensity values on the FTIR spectra were calculated using OMNIC 8.0 software (Spectra Tech, USA).

Flexural strength

Flexural strength was tested according to ISO 4049:2009 standard. PMMA resin specimens (2 mm × 2 mm × 25 mm, 12 specimens for each subgroup) were prepared in a rectangular stainless steel mold. After 24 h of setting, the specimens were removed from the mold. The three-point bending test was performed using a universal testing machine (AGS-10kNG, Shimadzu, Kyoto, Japan) at a cross-head speed of 1 mm/min, at a controlled room temperature. The flexural strength (FS) was calculated by the equations: FS=3PL2wh2

where P is the maximum load exerted on the specimen (N), L is the distance between the supports (25 mm), w and h are, respectively, the width (2 mm) and the thickness of the specimen (2 mm). Mean flexural strengths were calculated in MPa (megapascals). Three specimens were tested for each group. After testing, the microstructure of the fractured surfaces obtained from mechanical test were observed with SEM.

Microhardness

A microhardness test (Vickers test) was performed using the specimens after the three point bending test. Five half-bars from each subgroup were tested. Vickers microhardness was measured using a microhardness tester (HX-1000TM; Taiming, Shanghai, China). The indenter point was kept on the surface for 35 s with 25 g load. Three indentations were made on each specimen.

Surface roughness

Three half-bars from each subgroup tested by flexural strength testing were randomly chosen for analysis of surface roughness. The upper and lower surfaces of the specimens were polished with SiC grinding paper (320-grit, 600-grit, 1,200-grit and 2,000-grit) under running water on a polishing machine (UNIPOL-830; Shenyang Kejing Automation, Kejing, China). Specimens were ultrasonically (Vitasonic II; VITA Zahnfabrik, Berlin, Germany) cleaned for 10 min before testing. The surface roughness was analyzed with a 3D non-contact optical profilometer (PS50 Optical Profilometer; Nanovea, Irvine, California, USA). This device allowed a scanning over an area of 2 mm in length (x-axis) and 2 mm in width (y-axis). Surface roughness (Ra) was calculated for each scanned area.

Statistical analysis

The results for cell viability, degree of conversion, flexural strength, microhardness and surface roughness were analyzed using one-way analysis of variance (ANOVA), followed by Tukey’s test. SPSS 16.0 software (SPSS Inc, Chicago, Illinois, USA) was used for the statistical analysis. Statistical significance was preset at p = 0.05 for all tests.

Results

Quantification of NAC elution using HPLC

Figure 1 shows representative chromatograms for NAC standard solution (0.1 mg/mL) and eluted NAC from experimental specimen containing 0.9 wt% of NAC. Identification of the analyte, NAC, was made based on the retention time of the NAC peaks registered for the standard solutions. The relationship obtained for the linear peak area (A) and concentration (c) for the analyte (NAC) was: A = 27053c + 48133, with a correlation coefficient R2 = 0.99859. According to the calibration curve, the concentration of NAC released from the experimental resin containing 0.9 wt% of NAC was calculated as 0.54 ± 0.02 mM.

Figure 1 Typical HPLC chromatogram of standard solution of NAC (0.1 mg mL−1) and released NAC from experimental PMMA resin containing 0.9 wt% of NAC.

Identification of NAC was made based on the retention time of the NAC peaks registered for the standard solutions.

MTT assay

MTT assay revealed that, as compared to control group, cell viability at day 3 and day 7 was remarkably lower in the culture with PMMA extract (Fig. 2A). NAC exhibited protective effects on the cytotoxicity of PMMA in a concentration-dependent manner. Cultures with the extract of PMMA incorporating NAC showed higher cell viability than untreated PMMA resin. At day 7, relative cell viability was 77% for PMMA resin containing 0.15 wt.% NAC, which is more than 10-fold of that in subgroup 0 wt% (p < 0.05) (Fig. 2A). PMMA resin containing 0.9 wt% NAC presented the highest relative cell viability, reaching 95.6% at day 3 and 97.6% at days 7 (Fig. 2A). Figure 2B showed that 0.54 mM of NAC alone, based on results of HPLC, induced no significant influence on the viability of HDPCs (p > 0.05).

Figure 2 Cytotoxicity of PMMA resin with or without NAC.

Data are presented as the mean ± SD of three independent experiments performed in quintuplicates. (A) Effects of the extract of PMMA resin with or without NAC on the viability of HDPCs at day 3 and day 7. For each tested time period, values with different superscripts are significantly different from each other (One-way ANOVA, p < 0.05). (B) Effects of 0.54 mM NAC on the viability of HDPCs at day 3 and day 7. Values with different superscripts are significantly different from each other (One-way ANOVA, p < 0.05). NS, not significant between the control group and the experimental group. (C) Typical SEM pictures showing cell attachment and morphology on top of PMMA resins with or without NAC. The cells with round or collapsed appearances were observed in subgroups containing no or 0.15 wt.% NAC (arrows).

Cell attachment assay

Figure 2C shows the attachment of HDPCs on the surfaces of specimens at day one. In subgroup 0 wt% and subgroup 0.15 wt%, cells grew poorly. Cells with round or collapsed appearances were observed in these subgroups. In subgroup 0.3 wt%, 0.6 wt% and 0.9 wt%, HDPCs attached and spreaded well on the specimens, exhibiting spindle- or polygonal-shapes. The number of adhering cells increased as the concentration of NAC increased in the experimental PMMA resins. The surface of subgroup 0.9 wt% was almost fully covered by HDPC cells.

Degree of conversion

DC was measured from 5 min to 24 h after the liquid MMA was mixed with PMMA powder (Fig. 3). During the first 10 min curing period, DC increased dramatically. At 10 min, the DCs of the control and experimental subgroups ranged from 41.0% to 58.8%. Experimental subgroups containing NAC showed significant lower DC values as compared to control subgroup (p < 0.05). After 10 min, DC increased in a relatively slow manner. At 24 h, the DC values of the five experimental subgroups were all higher than 72.0%. In particular, the DC for experimental PMMA resin containing 0.15 wt% NAC reached 75.2% after 24 h of setting, showing no significant difference to the control PMMA resin without NAC (p > 0.05), whereas experimental PMMA resin with 0.3 wt% or higher concentration of NAC revealed slightly but statistically compromised DC values.

Figure 3 Degree of conversion of PMMA resin with different concentrations of NAC at specific time after mixing.

Results are presented as mean ± SD of three independent experiments.

Flexural strength testing

The flexural strength (FS) of specimens in various subgroups after 24 h of setting are shown in Fig. 4A. Adding NAC to PMMA resin induced a negative influence on the flexural strength of the material. Although the experimental resin containing 0.15 wt% NAC showed similar flexural strength (91.0 ± 6.2 MPa) to that of control group (95.9 ± 5.8 MPa), subgroups with higher concentration of NAC exhibited significantly compromised flexural strength, and the PMMA with 0.9 wt% showed the lowest flexural strength of 72.8 MPa (p < 0.05).

Figure 4 Mechanical properties of PMMA resin with or without NAC.

Data are presented as the mean ± SD. Values with different superscripts are significantly different from each other (One-way ANOVA, p < 0.05). (A) The flexural strength (FS) of specimens in various subgroups after 24 h (n = 12). (B) Microhardness value (VHN) after 24 h (n = 15). (C) Surface roughness (Ra) of the specimens (n = 3). (D) The SEM image of the fractured surfaces for each subgroup. Pit-like internal defects can be observed for subgroup with NAC (arrows).

Microhardness

Figure 4B showed the results of microhardness test after 24 h of setting. The trend of surface microhardness was similar to that of flexural strength. The incorporation of 0.15 wt% NAC had no significantly adverse effects on the microhardness of PMMA resin (p > 0.05 as compared to control group without NAC). However, when the concentration of incorporated NAC was further increased, the microhardness of the materials showed a significantly reduced value.

Surface roughness

Figure 4C show the mean Ra values of the specimens. The mean Ra values was 0.60 µm and 0.86 µm for subgroup 0.15 wt% and 0.3 wt%, respectively, which is not significantly different from that of control group. However, other subgroups with higher concentrations of NAC revealed significantly higher Ra values, indicating that adding NAC beyond the concentration of 0.3% has negative influences on the surface roughness of PMMA resin.

SEM of fractured surfaces

The SEM image of the fractured surfaces for each subgroup is presented in Fig. 4D. As can be seen, PMMA resin without NAC showed a relatively smooth appearance. The fractured surface of experimental resin containing 0.15 wt% NAC was similar to that of control group. However, for the subgroup with 0.3 wt% or higher concentration of NAC, pit-like internal defects can be clearly seen. Higher concentration of NAC incorporation is accompanied with greater number of internal defects.

Discussion

This study investigated the biocompatibility and mechanical properties of PMMA resin containing different concentrations of NAC. It was found that adding NAC to PMMA resin could reduce its cytotoxicity. However, the concentration of NAC should not exceed 0.15 wt% because too much NAC would compromise the mechanical and physical properties of the PMMA resin.

MTT study revealed that the exposure of HDPCs to eluent of PMMA resin resulted in an overwhelming cell death. Relative cell viability of HDPC cultured with eluent of PMMA resin was reduced to 12% at day 3 and 7.6% at day 7, indicating that this material is severely cytotoxic. This result is in accordance with previous studies (Att et al., 2009; Kojima et al., 2008; Tsukimura et al., 2009; Yamada et al., 2009; Yamada et al., 2008). It is well accepted that polymerization of resinous dental materials is never complete (Kopperud, Kleven & Wellendorf, 2011; Miletic, Santini & Trkulja, 2009) and the unpolymerized monomers can be released over time. The unpolymerized monomers can be the primary reason for the cytotoxicity of resinous dental materials (Att et al., 2009). In case of PMMA resin, its cytotoxicity can be attributed to the release of unpolymerized MMA monomer. Similar to other methacylate monomers, MMA can disturb the intracellular redox balance and thus induce negative influences on the function and viability of cells (Att et al., 2009; Ciapetti et al., 2002; Huang & Chang, 2005; Ratanasathien et al., 1995; Vale et al., 1997).

NAC, a well-known antioxidant, has been recognized as an effective agent that can reduce the cytotoxicity of dental monomers and thus improve the biocompatibility of resinous dental materials (Gillissen et al., 1996; Ma et al., 2013; Nocca et al., 2010; Paranjpe et al., 2007a; Paranjpe et al., 2007b; Paranjpe et al., 2009; Spagnuolo et al., 2013; Yamada et al., 2008; Zafarullah et al., 2003). In our study, it was found that incorporation of NAC to PMMA resin significantly salvaged HDPC from resin-induced cell death. When added at the concentration of 0.15 wt%, NAC remarkably increased relative cell viability by almost 10 times at day 7 (7.6% for control PMMA resin without NAC versus 77% for experimental resin containing 0.15 wt% NAC). This result is similar to that of Wael’s study, which found that cell density was significantly higher on NAC-supplemented resin compared to that on the untreated resin at days 2 and 5 (Att et al., 2009). To rule out the possibility that the improved cell viability related to NAC-containing PMMA resin is due to the pro-proliferating effects of NAC itself, we quantified the concentration of NAC released from the experimental PMMA resin containing 0.9 wt.% NAC and studied the influence of NAC perse on the proliferation of HDPC. According to the result of HPLC study, the concentration of NAC released was 0.54 mM. When 0.54 mM of NAC alone was used to treat the cells, no significant pro-proliferating effect was observed. To further prove the protective effect of NAC against PMMA-related cytotoxicity, we observed cell attachment on specimens from various subgroups. In accordance with the results of MTT study, SEM observation revealed that cell attachment and growth on PMMA resin containing NAC was remarkably enhanced as compared to the control group without NAC. Altogether, the results of MTT study and SEM observation indicate that incorporation of NAC can be an effective strategy to improve the biocompatibility of PMMA resin.

Aside from biocompatibility, the clinical success of the auto-polymerizing PMMA materials also depends on their mechanical properties (Balkenhol et al., 2008; Burke et al., 0000; Ferracane & Greener, 1986; Lovell et al., 2001; Lujan-Climent et al., 2008; van der Bilt et al., 2008). Thus, we investigated the influence of NAC incorporation on DC, flexural strength, microhardness and surface roughness of PMMA resin.

FTIR confirmed that the polymerization of PMMA resin with or without NAC was a time-associated reaction. Within the first ten minutes after mixing, the specimens polymerized at a relatively fast rate and the DC reached a high level after this period. This result is in agreement with previous studies, which found that the greatest increase of DC happens within the first 10 min after mixing of dental resins (Balkenhol et al., 2007; Ferracane, 1985). In our study, DC for the control group without NAC was as high as 58.8% after 10 min of setting. Incorporation of NAC reduced DC in a dose-dependent manner, with the subgroup containing 0.9 wt.% NAC presenting the lowest DC of approximately 41.0%. In the following period from 10 min to 24 h, the polymerization process of all subgroups slowed down and kept at an almost fixed rate. At 24 h, DC for all the tested subgroups was above 70%. While the experimental resin containing 0.15 wt.% NAC showed a DC value that is not significantly different from that of control, those incorporating higher concentration of NAC presented a significantly lower DC. Such reduced degree of conversion related to NAC-incorporating PMMA resin might be attributed to the creation of a heterogeneous system with reduced local concentration of MMA by unpolyerizable NAC.

Although DC is not the only factor that determines the mechanical properties of dental polymers, lower DC is usually associated with poorer mechanical properties. In our study, it was found that incorporation of NAC at concentrations higher than 0.15 wt.% significantly reduced flexural strength and microhardness of the PMMA resin. Observation of the fractured surface revealed that adding too much NAC produced defects in PMMA resin specimens. Such defects may represent voids that are left after the release of NAC or unpolymerized MMA. When such voids present on the outer surface of the material, rough surfaces may be produced. This explains the higher surface Ra values of resin specimens containing high concentration of NAC.

Although too much NAC imposed negative influences on flexural strength, microhardness and surface roughness of the carrier material, limiting the concentration of NAC to 0.15 wt% can be an effective strategy to generate experimental PMMA resin with acceptable mechanical properties. As for flexural strength, the measured results of PMMA incorporating 0.15 wt.% and 0.30 wt.% NAC were 91.0 MPa and 83.3 MPa, respectively, which all exceeded the minimum values stipulated by the ISO flexural strength standard of 80 MPa for self-curing dental polymer-based restorative materials. Although there is no specific requirement for microhardness in the ISO standard, the value of our experimental PMMA resin containing 0.15 wt.% NAC was comparable to other commercial resin materials for temporary restorations (Jo, Shenoy & Shetty, 2011). As for surface roughness, the Ra value for 0.15 wt.% NAC-containing PMMA resin was 0.6. According to some studies, surfaces with Ra values lower than 0.75 µm can be classified as smooth surfaces (Busscher et al., 1984; Quirynen et al., 1990). Moreover, the Ra value of 0.6 µm for PMMA resin containing 0.15 wt.% NAC is close to the recommended surface roughness threshold (0.2 µm) to minimize plaque accumulation (Bollenl, Lambrechts & Quirynen, 1997; Quirynen et al., 1990; Ulusoy, Ulusoy & Aydin, 1986; Verran & Maryan, 1997).

To conclude, incorporation of NAC reduced the cytotoxicity of dental PMMA resin. Although adding too much NAC to PMMA resin resulted in compromised DC, flexural strength, microhardness and surface roughness, limiting the concentration of NAC within 0.15 wt.% remarkably improved biocompatibility of PMMA resin without exerting significant negative influence on mechanical properties. Therefore, incorporation of NAC should be considered as an effective strategy to produce clinically reliable and biocompatible PMMA dental resins.

Supplemental Information

Supplemental Information 1 Supplemental Files

Click here for additional data file.

Additional Information and Declarations

Competing Interests

Author Contributions

Human Ethics

The authors declare there are no competing interests.

Yang Jiao performed the experiments, analyzed the data, wrote the paper, prepared figures and/or tables, reviewed drafts of the paper.

Sai Ma performed the experiments, analyzed the data, contributed reagents/materials/analysis tools, wrote the paper, reviewed drafts of the paper.

Jing Li and Lequn Shan contributed reagents/materials/analysis tools, reviewed drafts of the paper.

Yanwei Yang and Meng Li contributed reagents/materials/analysis tools, prepared figures and/or tables.

Jihua Chen conceived and designed the experiments, wrote the paper, reviewed drafts of the paper.

The following information was supplied relating to ethical approvals (i.e., approving body and any reference numbers):

In our study, human dental pulp cells (HDPCs) were isolated from dental pulp tissue of non-carious third molars extracted from young healthy patients (18–25 years old) according to a protocol verbally approved by the Ethics Committee of the Fourth Military Medical University.

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
