# Peer review of "The influences of N-acetyl cysteine (NAC) on the cytotoxicity and mechanical properties of Poly-methylmethacrylate (PMMA)-based dental resin"

_PeerJ, doi:10.7717/peerj.868_

## Round 0.1 · original submission · Minor Revisions

Based on my own reading, in addition to the reviewer's comments I would also like to ask for a few alterations to the manuscript:

1. In the introduction GSH is described as an enzyme when it is just a peptide. Please cane you rephrase line 53.
2. In the Cell viability section of th ematerials and methods line 122 did you mean sterile filtered instead of disintected by infiltration?
3. I think table 1 would be clearer as a line graph to show the temporal change with degree of conversion.
4. The use of letters for symbolizing the degree of significance needs careful and further explanation, either in the methods section on statistical analysis or in the figure legends describing the graphs where it appears. For instance A=p<0.05 etc. It should be noted which comparison is being displayed eg all data compared to control (no PMMA).

Reviewer 1 ·

Basic reporting

The manuscript is written in a proper way and clearly presents the background literature, the aim of the study and the design of the experiments. The Figures and the Table are presented in a proper way.

Experimental design

PMMA based dental resin is one of the most frequently used materials in dentistry. However, un-polymerized monomers released from the resin are cytotoxic molecules because they disturb the intracellular redox balance. Therefore, it is suggested to incorporate NAC in PMMA resins in order to protect the cells from oxidative stress induced by the PMMA monomers.
The current study investigates the influences of NAC on cell survival and attachment as well as on important mechanical properties of PMMA resin. The experiments are well designed and performed in a proper way.
Technical comments:
Fig. 1: How many times the determination of extracted NAC concentration was repeated?
Line 226: p > 0.05 probably should be p< 0.05
Line 229: p < 0.05 probably should be p> 0.05

Validity of the findings

It is concluded that NAC (0.15%) remarkably improved biocompatibility of PMMA resin without exerting significant negative influence on its mechanical and physical properties.
The Figures and the Table supports the conclusion presented in the manuscript.

---

## Round 0.2 · accepted · Accept

Thank you for your changes to the manuscript. I have no further comments or changes to make.